# Evaluation of *Dittrichia viscosa* Aquaporin *Nip1.1* Gene as Marker for Arsenic-Tolerant Plant Selection

**DOI:** 10.3390/plants11151968

**Published:** 2022-07-28

**Authors:** Angelo De Paolis, Monica De Caroli, Makarena Rojas, Lorenzo Maria Curci, Gabriella Piro, Gian-Pietro Di Sansebastiano

**Affiliations:** 1Institute of Sciences of Food Production (ISPA-CNR), 73100 Lecce, Italy; angelo.depaolis@ispa.cnr.it; 2DiSTeBA (Department of Biological and Environmental Sciences and Technologies), University of Salento, 73100 Lecce, Italy; monica.decaroli@unisalento.it (M.D.C.); makarena.rojas@unisalento.it (M.R.); lorenzomaria.curci@studenti.unisalento.it (L.M.C.); gabriella.piro@unisalento.it (G.P.)

**Keywords:** *Dittrichia viscosa*, aquaporin, *Nip1.1*, arsenic, phytoremediation, real-time PCR, selection marker, genetic improvement

## Abstract

*Dittrichia viscosa* (L.) Greuter is gaining attention for its high genetic plasticity and ability to adapt to adverse environmental conditions, including heavy metal and metalloid pollution. Uptake and translocation of cadmium, copper, iron, nickel, lead, and zinc to the shoots have been characterized, but its performance with arsenic is less known and sometimes contradictory. Tolerance to As is not related to a reduced uptake, but the null mutation of the aquaporin *Nip1.1* gene in Arabidopsis makes the plant completely resistant to the metalloid. This aquaporin, localized in the endoplasmic reticulum, is responsible for arsenite and antimony (Sb) membrane permeation, but the uptake of arsenite occurs also in the null mutant, suggesting a more sophisticated action mechanism than direct uptake. In this study, the *DvNip1* gene homologue is cloned and its expression profile in roots and shoots is characterized in different arsenic stress conditions. The use of clonal lines allowed to evidence that *DvNip1.1* expression level is influenced by arsenic stress. The proportion of gene expression in roots and shoots can be used to generate an index that appears to be a promising putative selection marker to predict arsenic-resistant lines of *Dittrichia viscosa* plants.

## 1. Introduction

*Dittrichia viscosa* (L.) Greuter, also known as woody fleabane [1], gained attention for its high genetic plasticity and ability to adapt to a wide range of environmental conditions including heavily polluted mining or industrial sites and arid or high salinity land. *D. viscosa* was found able to uptake and translocate cadmium (Cd), copper (Cu), iron (Fe), nickel (Ni), lead (Pb), and zinc (Zn) to the shoots [2,3,4,5]. *D. viscosa* is not a hyperaccumulator plant but can grow in high drought conditions while still producing large biomass, even tolerating significant concentrations of As[III], As[V], and Cd[II]. In spite of these remarkable characteristics, adaptive modification obtained in wild populations are not ideal characteristics when applied to phytoremediation in which predictable results are expected [5,6]. Genetic stability is a desirable trait and interest in the genetic improvement of this plant species is increasing.

The interest in *D. viscosa* is particularly related to the potential adaptation of wild populations to tolerate arsenic. In addition to the natural arsenic pollution of soils, the application of fertilizers, pesticides, desiccants [7], and growth promoters for animals [8] containing arsenic elements has led to widespread As pollution of soils. With increasing concentration, high As concentration becomes toxic for all plants, causing chlorosis, necrosis, inhibition of growth, and finally death. Uptake of As by plants occurs primarily through the root, and until now it is not well known.

*D. viscosa* uptake and translocation of Cd, Cu, Fe, Ni, Pb, and Zn to the shoots has been characterized [2,3,9] but its performance with arsenic (As) is less understood [6,10] and some data are contradictory. One study reported As to be fully translocated to *D. viscosa* shoots and volatilized [6], while another detected As stabilized in the roots [10]. Regardless, there is substantial variability among plants. In a previous work we studied the accumulation of As[III] and As[V], confirming the high variability among individuals and establishing a clonal population to promote functional genetic investigations [5].

As a phosphate analog, As[V] enters plant cells through phosphate transporters (PHTs), while As[III] and methylated As species enter the cells and are translocated or detoxicated by aquaglyceroporins and several additional transporters [11]. Despite this complex scenario, it was shown that tolerance to As is not related to a reduced uptake, but it is increased by the mutation of the aquaporin *AtNip1.1* [12,13]. The null mutant of the gene in Arabidopsis makes the plant resistant to the metalloid. This aquaporin is responsible for arsenite (As[III]) [12] and antimony (Sb) [14] membrane permeation, but the uptake of As[III] also in the null mutant [13,15] suggests a more sophisticated mode of action.

Aquaporins (AQPs) are tetrameric integral membrane proteins forming a channel to facilitate the movement of water and small molecules. Their sequences are highly conserved across plant species because they have essential roles, not only in regulating osmotic balance but also in transporting non-ionic, small neutral solutes and ions, such as nitrate, chloride, and ammonia [16]. They have six transmembrane (TM) alpha helices and two domains responsible for selectivity: the NPA domain (asparagine—proline—alanine) and the ar/R (aromatic arginine). These domains insert from opposite sides of the membrane and constrict the center of the pore, forming a filter for solute selectivity [17,18,19].

Plant AQPs are phylogenetically classified in five major subfamilies: plasma membrane intrinsic proteins (Pip), nodulin 26-like intrinsic proteins (Nips), tonoplast intrinsic proteins (Tips), small intrinsic proteins (Sips), and uncharacterized X intrinsic proteins (Xips) [20]. The sub-cellular localization pattern of plant AQPs is complex and does not correspond to subfamilies. Studies in tobacco leaves revealed a dual localization of the Pip NtAQP1 (homologue of AtPip1.2) in the plasma membrane and inner chloroplast membrane [21]. Tips, initially thought to be vacuolar, were found in multiple sub-cellular localizations. Tagged variants of AtTip3.1 and AtTip3.2 labelled both tonoplast and plasma membrane (PM) of Arabidopsis embryo cells; the pollen-specific AtTip5.1, tagged with GFP, was found to co-localize with the mitochondrial marker Mitotracker in transgenic vegetative tissues. Tips isoforms are markers able to label morphologically and functionally distinct vacuoles [22], implying that each Tip isoform has to be differentially sorted.

The aquaglyceroporin AtNip1.1 (nodulin 26-like intrinsic protein 1.1), initially expected to be found at the PM [12], was later localized at the ER and in part on the tonoplast, where it was shown to interact with the SNARE AtSYP51 [13]. AQP interaction with SNAREs may represent an important regulatory mechanism for membrane traffic (Barozzi et al., 2019). In fact, more examples of interaction exist: Pip2.5 was found to interact physically with SYP121 [23] and Pip2.7 with SYP121 and SYP61 [24].

In addition to facilitating water movement and the role in membrane organization, possibly in the traffic events, it should be noted that other small molecules cross the membrane within these molecular channels. These may be elements essential to life but also pollutants with toxic effects such as heavy metals and metalloids (from now on all are indicated as HM).

*A. thaliana* Nip1.1 showed to be important for membrane direct traffic from ER to tonoplast [13,25]. Its localization in the ER membrane suggests it is not essential for As[III] uptake. In fact, in two different studies [13,15] it was shown that null mutant *nip1;1* ko lines were able to uptake As[III] anyway. This is not surprising, because it is known that As[III] uptake may rely on multiple mechanisms. In rice and Arabidopsis roots the uptake of As[III] is known to occur via two different mechanisms [26]. If *nip1;1* ko has no effect on As(III) uptake, but the mutant is nonetheless resistant to the metalloid, then translocation or compartmentalization of the metalloid may be involved in a tolerance mechanism.

In this study, the *DvNip1* gene homologue was isolated in *D. viscosa* (L.) Greuter. The expression analysis of *DvNip1.1* in clonal lines of *D. viscosa* grown in different As stress conditions were also reported, and the ratio of *DvNip1.1* expression levels in roots and shoots is proposed as a putative selection marker to identify highly resistant lines of *D. viscosa* plants.

## 2. Results

### 2.1. Cloning of Dittrichia Viscosa Nip1.1 Gene

In order to isolate *D. viscosa* putative *DvNip1.1*, a genomic fragment of about 700 bp was amplified on leaf genomic DNA using degenerate primers designed on conserved amino acids of the NPA domain (asparagine–proline–alanine) of plant aquaporins genes already sequenced. The 700 bp sequence was analyzed to find putative splice sites by using the FSPLICE program and an in silico coding sequence was identified after exon junction. A sequence of 115 amino acids belonging to *DvNip*, between the NPA domains, was compared with those of plant AQPs using the program Blast-P. A high (>96%) identity was observed with aquaporin Nip1.1 from *Helianthus annuus*, sequence ID: XP_021979651.1., an Asteraceae plant species. A significant amino acid sequence similarity with other Nip1 members reported in databases was detected.

To further isolate the full-length coding sequence of *DvNip* members, gene-specific primers and the 3′ or 5′ universal primer of the RACE kit were used as described in Materials and Methods. This approach enabled us to obtain a full-length ORF of a new *Dittrichia viscosa* Nip1 member submitted to the NCBI database (BankIt2605196 BSeq#1 OP046365) here designated *DvNip1.1*. The full-length cDNA sequence of *DvNip1.1* contains 1014 nucleotides with a 63-bp 5′ UTR, an 834-bp ORF, and a 117-bp 3′-UTR (Figure 1). *DvNip1.1* encoded a polypeptide containing 277 amino acid residues with a predicted theoretical molecular mass of molecular weight 29.36 kD and an isoelectric point of 8.90 as calculated using the ExPASy-Bioinformatics Resource Portal. A phylogenetic analysis of Nips1 of different plant species showed that the DvNip1 member, here identified, clustered with the Nips1 belonging to other Asteraceae plant species such as >PWA69246.1 (*Artemisia annua*); >XP_023743033.1 (*Lactuca sativa*); >XP_024983052.1 (*Cynara cardunculus* var. scolymus); >XP_021979651.1 (*Helianthus annuus*) (Figure 2).

### 2.2. Identification and Validation of Putative Reference Gene in Dittrichia Viscosa

To identify stable reference genes in *D. viscosa* subjected to various heavy metal stresses, we tested the possibility of using Actin 2, Actin 8, and EF1-A. We used primers previously used in Arabidopsis for amplification of *AtAct2* (AT3G18780), *AtAct8* (AT1G49240), and *AtEF1-A* (AT1G18070) and in tobacco for amplification of *NtEF1-A* (AF120093) (Table 1). All primer couples were tested under all the applied experimental conditions, both on aerial parts and root-derived mRNA samples.

The primers designed on *AtAct8* did not give amplification products, therefore the analyses continued with the other three primer pairs. The size of PCR products for each reference gene was visualized by electrophoresis on 2.0% agarose gel, and confirmed to produce a single amplicon (Appendix A). The specificity of each primers pair was verified by melting curve analysis, which revealed that each gene had a single amplification peak. Means and standard deviation of the Ct values for each putative reference genes were analyzed in all samples for all experimental conditions and compared (Figure 3). The results of the reference genes’ amplification displayed a wide range of transcription levels across all analyzed samples, with average Ct values ranging from 23 to 35 (Appendix A). Since gene expression levels are negatively correlated with Ct values, the amplification product obtained with *AtEF1-A* primer pairs exhibited the highest abundance, with the lowest mean Ct values, 23.77 ± 0.64 (shoots)/24.99 ± 0.98 (roots). The amplimers obtained with the *AtAct2* and *NtEF1-A* couple of primers showed relatively low expression, with a mean Ct value of 30.19 ± 0.10 (shoots)/31.25 ± 0.45 (roots) and 36.43 ± 1.49 (shoots)/34.94 ± 1.75 (roots), respectively.

In addition to the abundance of product amplified by the *AtEF1-A* primer pair, it also showed the lowest coefficient of variance (CV) among those tested (Appendix A). All the data indicated that the *AtEF1-A* primer pair was eligible for the amplification of an appropriate reference gene for *Dittrichia* in all samples and experimental conditions of this research.

The amplimer obtained with the *AtEF1-A* primer pair on leaf cDNA was checked by sequencing (Eurofins Genomics, Ebersberg, Germany). The deduced amino acid sequence (Expasy Translate Tool) was analyzed through the Basic Local Alignment Search Tool (BLASTP) with the other Asteraceae species identified with phylogenetic analysis (*Artemisia annua*; *Lactuca sativa*; *Cynara cardunculus* var. scolymus; *Helianthus annuus*; Figure 2). The results confirmed the specificity of primers, since the deduced amino acid sequence showed 100% of identity with the E value of 1e-13 with the elongation factor 1-Alpha of all queried species (Appendix A). On the *EF1-A* nucleotide sequence of *Helianthus annuus* (GeneID:110868526), a primer pair was drawn upstream and downstream of the identified partial nucleotide sequence in order to clone a wider sequence of putative *EF1-A* gene in *Dittrichia* (Table 1).

A fragment of 675 bp was amplified and the deduced sequence of 225 amino acids was compared with those known of EF1-A using the program Blast-P. A high (99%) identity was observed with *EF1-A* from *Helianthus annuus*, sequence ID: XP_021973407.1. The obtained partial sequence of *DvEF1-A* was submitted to the NCBI database (BankIt2605382 BSeq#1 OP047689).

### 2.3. Real-Time Analysis of DvNip1.1 Expression

*DvNip1.1* gene expression level has been evaluated in the presence of As[III], As[V], and Cd[II] to investigate its role in arsenic absorption. Studies were conducted on DI3-F plants, a clonal population composed of plants with the same genetic heritage, allowing us to evaluate, through replicas, the real effect of each treatment on *DvNip1.1* expression. As[III] was added (90 µM) in the medium of hydroponic culture and RNA was extracted after 6 and 48 h to test response intensity and persistence in time. As[V] (90 µM) and Cd[II] (100 µM) were tested only for 6 h to verify the specificity of response to As[III]. *DvNip1.1* was found downregulated by As[III] and As[V], but not by Cd[II].

In roots, a 4-fold downregulation of *DvNip1.1* was observed after 6 h, while in shoots there was not a relevant variation. Similarly, the effect of As[V] induced a 3-fold downregulation in roots but not in shoots. At 48 h after treatment, *DvNip1.1*’s effect was still detectable but reduced to a 3-fold downregulation. No expression modification was induced by Cd[II], confirming a specific relation of *DvNip1.1* with As. No significant variation in the expression of *DvNip1.1* was observed in the shoots during HM treatments (Figure 4).

### 2.4. DvNip1.1 Expression and As Tolerance

Plants derived from the wild population DI3 segregated on 50 µM As[III] were analyzed to test if NIP1.1 expression was correlated to tolerant phenotype. To test the hypothesis, two groups of five plants each were selected among the plants that showed the weakest (weak plants) or the strongest phenotype (strong plants) while growing on As[III] in term of different sizes and chlorosis. Clonal individuals of these plants were propagated as previously described [5] and transferred in hydroponic culture similarly to DI3-F plants used to evaluate expression levels. The *DvNip1.1* gene expression was tested by RT-qPCR without treatment. Since in this case no control was available, we established an index to avoid the heterogeneity related to genetic variability. The index corresponds to the rate between expression in the root and in aerial parts. The five independent clones derived from originally “weak” plants showed an index within the range of 0.6 and 0.8; the five independent clones derived from originally “strong” plants showed an index within the range of 0.18 and 0.42. Both selected groups of clones had a much lower expression of *Nip1.1* in the root, and as a consequence, lower index than DI3-F. The difference between the two groups was highly significant (Figure 5).

## 3. Discussion

Soil decontamination by physical-chemical technologies is possible but extremely expensive. These practices may lead to serious changes in soil properties, being often destructive and/or the cause of releasing additional contaminants to the environment [28]. Phytoremediation through phytoextraction [29,30] appears to be an eco-friendly and much cheaper solution, especially when considering the capacities of hyperaccumulator plants [31]. Unfortunately, hyperaccumulator plants may have small biomass or require specific cultural cares. *D. viscosa* is not a hyperaccumulator plant but can grow in stress conditions, producing large biomass in the presence of As[III], As[V], and Cd[II]. The HMs or metalloids accumulated may reach significant concentrations. Interestingly, the bioconcentration capacity for As is inversely correlated with As concentration in the soil [5]. With an As[III] concentration of 15 µM, the bioconcentration average is 5.3, while in presence of 45 µM it is 10-fold reduced to 0.53. Inverted correlation is replicated with As[V]. On the contrary, the bioconcentration average for Cd[II] is directly proportional to concentration in soil [5].

Selected *D. viscosa* clones, exhibiting differentiated behaviors in terms of tolerance to arsenic, might be potentially used for phytoremediation of As-polluted soils with different approaches. The plants may be more tolerant and very effective for re-naturalization of polluted areas or able to uptake more contaminants suffering more intoxication but performing better phytoextraction. The knowledge about the exact proportion between As uptake and biomass becomes relevant for phytoremediation applications. This analysis requires the use of stabilized cultivars, eventually of clonal origin. By producing these cultivars, we have the occasion to perform genetic improvement, but the acceleration of the process requires the availability of valid selection markers.

Here we demonstrate that the expression levels in *D. viscosa* of the aquaporin gene homologous to the *Arabidopsis thaliana* aquaporin *Nip1*.1 allows the prediction of As tolerance and, as a consequence, the potential phytoremediation efficiency.

Aquaporins evolved modifying both specificity of their expression pattern and water permeation properties, maintaining the ability to be differentially induced. Therefore, certain expression of maize AQPs correlates with nitrate, sulfate, auxin, sucrose, mercury, chloride transmembrane transporters, and metal ion transporters such as iron, potassium, copper, and cobalt [32]. In the case of Nip1.1, overexpression appears to enhance Zn[II] uptake but also increases As[III] translocation to aerial parts. Translocation relies on intracellular homeostasis and cell-to-cell communication through the ER. Nip1.1 may increase translocation of As[III], coherently with its localization in the ER [13], but it has a prominent role in intracellular compartmentalization so that its absence alters the vacuole characteristics and, as a consequence, As[III] tolerance. Re-localization and post-translational modification of AQPs could be involved not only in reducing water flow and HM permeation but also in adjusting cellular osmotic potential and specific compartmentalization in response to stress [33,34].

We cloned a *D. viscosa Nip1.1* homologue showing >96% identity with aquaporin *Nip1.1* from *Helianthus annuus* (ID: XP_021979651.1). It was interesting to monitor the expression of this gene with Real-time quantitative PCR (RT-qPCR) to interpret the behavior of this plant in the presence of arsenic. RT-qPCR is a powerful tool for precise analysis of gene expression levels in different samples and under different experimental conditions. To use this technique, it is well known that selecting the appropriate reference genes is crucial to the accuracy and reliability of RT-qPCR data. The ideal reference gene should have a moderate and stable expression level in different tissues at different developmental stages and under different experimental conditions. Among the most frequently used reference genes in plants for RT-qPCR are 18S rRNA, Actin (*Act2* and *Act8*), α-tubulin and β-tubulin (*TUA* and *TUB*), and elongation factor-1A (*EF1-A*) [35,36,37]. Since no reference genes were available for transcript normalization in *D. viscosa*, we first sought to identify stable reference genes in *D. viscosa* subjected to various heavy metal stresses. We found that primers designed on the *AtEF1-A* (AT1G18070) sequence amplify a very good reference gene that, partially cloned from *D. viscosa* (675 bp), showed high identity with *EF1-A* from *Helianthus annuus* as predicted by the phylogenetic analysis of plant *Nips1*.

RT-qPCR experiments, during the treatment of plants in hydroponic culture with As[III], As[V], and Cd[II], demonstrated that the immediate effect of the pollutant was detected only in roots, and it was specific for As[III] and As[V] within 6 h. After 48 h, the downregulation of *Nip1.1* expression persisted but in a reduced proportion. Cd[II] treatment, at toxic doses, does not alter *Nip1.1* expression. The observed response specificity for As but not for Cd excludes the possibility that non-specific stress may downregulate *Nip1.1*, as observed in other studies [38].

We can conclude that this AQP is part of a specific short-term response to arsenic intoxication. The expression downregulation in the root is not stabilized in longer periods and is not extended to the aerial part of the plant, nonetheless, a long-term effect is often observed in the progeny of plants grown under stress.

To verify the possibility to correlate *Nip1.1* expression and tolerance to As, we screened plantlets derived from the wild population DI3 grown on As[III] 50 µM. When fitness differences were more evident, with different sizes and chlorosis, two groups of plants were rescued and moved in vitro for further clonal propagation in optimal conditions. One group of plants was selected for their strong appearance during the treatment and presumably their higher tolerance to As (strong plants). The second group, conversely, included plants severely affected by the presence of As (weak plants). To characterize the expression level of *Nip1.1* in these two groups of plants, in the absence of As, we established an index to avoid the heterogeneity related to genetic variability. The index corresponds to the rate between expression in the root and in aerial parts. In several plant species, *Nip1.1* is generally expressed in roots more than in shoots [39]; as a consequence, downregulation of its expression produces an index reduction. In control conditions, in the absence of HMs, the index measured in plants that survived an As treatment appeared lower than in control. The plants with the higher tolerance to As (strong plants) had a significantly lower expression level of *Nip1.1* already in control conditions. This experiment supported our hypothesis that tolerance to As may be related to a basic low expression level of *Nip1.1*.

## 4. Conclusions

These observations are important to identify a marker for As tolerance considering that other channels and transporters are related to As and its uptake. Even among AQPs, AtNip5.1, OsPIP2.4, OsPIP2.6 and OsPIP2.7 have all been shown to influence As[III] uptake and tolerance [11].

We can consider *DvNip1.1* expression rate between root and shoot as an index of As tolerance and use this knowledge to assist the selection of plants with increased resistance. This is particularly interesting when considering the wide range of applications of *D. viscosa* cultivation for phytoremediation or for producing bioactive compounds. At the same time, the role of *Nip1.1* appears to be conserved, and the possibility of genetic improvement can be extended to other plant species.

## 5. Materials and Methods

### 5.1. Plant Material

All plants used in the present study were derived by in vitro propagation of a single plant obtained from population DI3 present at Campus Ecotekne (40 20′01.8″ N 18 07′23.3″ E). The original plant survived a selection on 100 µM As[V] described by Papadia and co-workers [5]. The clonal line was named DI3-F and propagated in solid Murashige and Skoog medium (MS) basal medium.

### 5.2. DNA and RNA Extraction

Plant tissues for nucleic acid extraction were obtained from *D. viscosa* L. plants in vitro. Genomic DNA was extracted using a DNeasy Plant Mini Kit (Qiagen, Milan, Italy). Total RNA was isolated using the SV Total RNA Isolation System (Promega, s.r.l. Milan, Italy). The RNA samples were treated with RQ1 RNase-Free DNase (Promega, Milano, Italy). Nucleic acid concentrations were determined spectrophotometrically, and the quality was checked by agarose gel electrophoresis.

### 5.3. Isolation of Sequences Encoding Partial Putative DvNip1 Gene

An amount of 100 ng of genomic DNA was used as a template, in the presence of 20 pmol of each primer (Table 1), 0.2 mM of each dNTP, 2.5 U of Polymerase Mix (thermo fisher) in 50 μL of the buffer solution, as indicated by the supplier. After a denaturation step at 94 °C for 3 min, amplification reactions were carried out for 35 cycles for 1 min at 94 °C, 1 min at the annealing temperature of 50 °C, and 1 min at 72 °C. A final elongation step was run at 72 °C for 15 min. The amplification products were cloned into pGEM-T Easy vector (Promega s.r.l., Milan, Italy).

### 5.4. cDNA Synthesis and Amplification

Three micrograms of total RNA were treated with RQ1 RNase-Free DNase (Promega, Milano, Italy) and the cDNAs were synthesized using random primers and the GoScript™ Reverse Transcription System (Promega), according to the manufacturer’s instructions. Reverse Transcription-PCR cycles were as follows: 25 °C for 5 min, 42 °C for 1 h, and 70 °C for 15 min. The PCR amplification was performed using the following conditions: 3 min at 94 °C followed by 35 cycles at 94 °C for 30″, at 52 °C for 30″, at 72 °C for 1 min, and a final 10 min at 72 °C. After agarose gel electrophoresis, the amplified fragment was cloned into the pGEM-T Easy vector (Promega s.r.l., Milan, Italy).

### 5.5. Cloning and Sequence Analysis

In order to clone *D. viscosa* putative Nip gene family members, degenerate primers DvNipDegR and DvNipDegF (Table 1) were designed based on the NPA domain conserved in protein sequences of the Nip family from *Arabidopsis thaliana* and other Asteraceae plants already published (Appendix A). Then, the genomic region between the two NPA domains was amplified using polymerase chain reaction (PCR). Subsequently, based on the sequence information of the genomic DNA fragment, the 3′ and 5′ region were isolated using specific primers and rapid amplification of cDNA ends using the 3′ RACE System (Life Technologies Ltd., Paisley, UK). DvNip3′GSP1-F/RACE Abridged Universal Amplification Primer (AUAP) was used for the isolation of the 3′ region and DvNip5′GSP1-R/nested DvNip5′GSP2-R and RACE (AUAP) were used for the 5′ region of Nip1 according to the manufacturer’s instructions. Finally, a pair of gene-specific primers, DvNip-Full-F and DvNip-Full-R (Table 1), were designed and used to amplify the full-length sequence of the DvNip1 gene. The amplified product was then cloned into pGEM-T Easy vector (Promega Italia s.r.l., Milano) and used to transform competent *E. coli* strain XL1-Blue (Stratagene, La Jolla, CA, USA) using standard procedures. Recombinant DNA plasmids were extracted using a Qiagen plasmid midi Kit (Qiagen, Milano, Italia) and the nucleotide sequences were obtained using an ABI 3130 DNA Sequencer (Applied Biosystems, Foster City, CA, USA). Sequence analysis and comparison with other aquaporin sequences were performed using the NCBI BLAST web tool (https://blast.ncbi.nlm.nih.gov, accessed on 22 May 2021) (Appendix A). The FSPLICE program (http://www.softberry.com, accessed on 15 June 2021) was used to identify the intron/exon sequences. The molecular weight and the theoretical pIs was predicted using the ProtParam tool (https://www.expasy.org/protparam/, accessed on 15 October 2021). Alignment of deduced amino acid sequences of *DvNip1* with other *Nip* encoding sequences was obtained using the program MultAlin, created by Florence Corpet, CNRS Toulouse, France (http://multalin.toulouse.inra.fr/multalin/accessed on 15 October 2021) [40]. A phylogenetic neighbor-joining analysis was conducted on deduced amino acid sequences using the program MEGA version 6.0 (www.megasoftware.net accessed on 15 October 2021) [41]. The program DeepTHMM (https://dtu.biolib.com/DeepTMHMM accessed on 15 October 2021) was used for transmembrane (TM) prediction [42].

### 5.6. Hydroponic Culture and HM Treatments

Hydroponic culture of DI3-F was established to provide the conditions to perform the HM stress tests. Four plant tips of D13-F lines in vitro were transferred into holes of a plastic grid in sterile boxes containing one hundred milliliters of liquid MS basal medium. After fifteen days of growth in these conditions, several roots were developed, ranging from 3 to 12 cm each. For the HM treatment, a single element was added to the liquid medium: As[III] (NaAsO_2_) up to 90 µM, As[V] (Na_2_(AsHO_4_) 7H_2_O) up to 90 µM, or Cd[II] (Cd(NO_3_)_2_) up to 100 µM. The experiment was carried out at 23 °C, for 6 and 48 h in the case of As[III] and 48 h in the case of As[V] or Cd[II]. Immediately after, the samples were collected; shoots and roots were separated carefully cutting at the root crown, frozen with liquid nitrogen, and stored at −80 °C before RNA extraction.

Three-week-old plantlets from the wild population DI3 cultivated in vitro in solid MS medium (2.2 g/L MS salts including vitamins from Duchefa-Biochemie, 1% sucrose) were transferred on Petri dishes containing the same media supplemented with As[III] 50 µM for three weeks. The selection of “weak” and “strong” plants was based on their fitness, size, and chlorosis. Selected plants were kept in MS medium for 4 weeks for growing and finally transplanted into pots. Weak and strong plants selected for tolerance to As[III] were propagated into clonal lines (W1-4, S1-4) as previously described [5], and adapted to hydroponic cultures similarly to DI3-F.

### 5.7. RNA Extraction and Real-Time Analysis

Total RNA was isolated from the roots and shoots of plants subjected or not to HM treatments as previously reported [37]. RT-qPCR analysis was performed using SYBR Green fluorescent detection in a CFX96 Real-Time System Cycler (Bio-Rad, Hercules, California, United States) with three biological and three technical replicates per sample. The primer sequences used are reported in Table 1. The PCR program was as follows: 10 min at 95 °C, 50 cycles of 15 s at 95 °C, 20 s at 60 °C, and an increment of 0.5 °C every 0.5 s from 65 °C to 95 °C. The specificity of PCR products was checked in a melting curve test. Differences in gene expression between treated and untreated samples were considered significant when the expression was at least doubled (greater than or equal to two-fold upregulation) or halved (less than or equal to two-fold downregulation) according to [43].

### 5.8. Selection of Candidate Reference Genes

In the absence of *D. viscosa* genomic information, primers used for amplification of *Arabidopsis Actin* 2 (*AtAct2*, AT3G18780), *Actin 8* (*AtAct8*, AT1G49240), elongation factor 1A (*AtEF1A*, AT1G18070), and *Nicotiana tabacum* (*NtEF1-A*, AF120093) genes were tested as reference genes (Table 1). The *AtEF1-A* primer was chosen to amplify the reference gene because it showed the lowest variation coefficient compared to other primers. The amplimer obtained with the primer pair for *AtEF1-A* had a variation coefficient below 0.1, which is described as negligible variation under treatments in *Arabidopsis*, according to [35]. The obtained amplimer was cloned in pGEM-T Easy (Promega s.r.l., Milan, Italy) and a single clone was checked by sequencing (Eurofins Genomics, Ebersberg, Germany). Then, a wider, albeit partial, sequence of a putative *DvEF1-A* gene was obtained with a primer pair drawn on *HaEF1-A* (Table 1).

## Figures and Tables

**Figure 1 plants-11-01968-f001:**
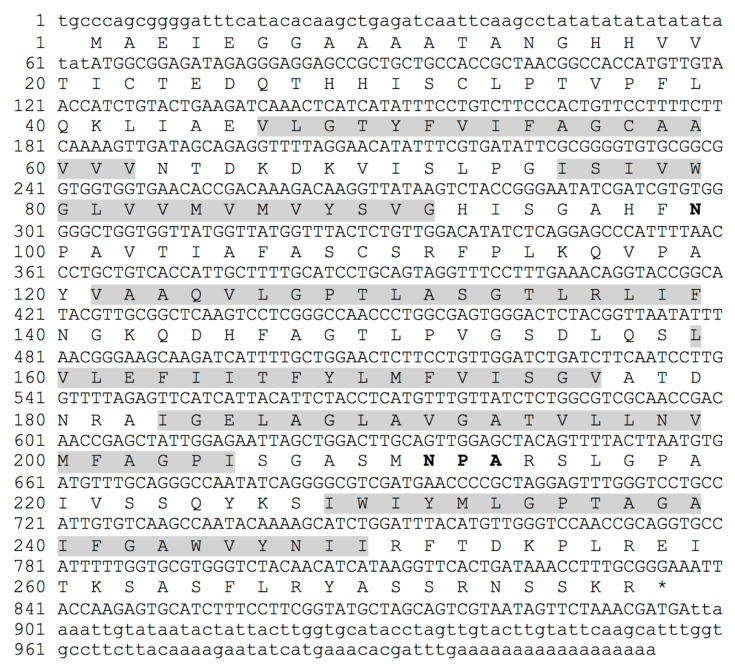
Nucleotide and deduced amino acid sequences of DvNip1 gene. NPA domain sequence is indicated with bold font for amino acids. Transmembrane TM1/TM6 are in grey background.

**Figure 2 plants-11-01968-f002:**
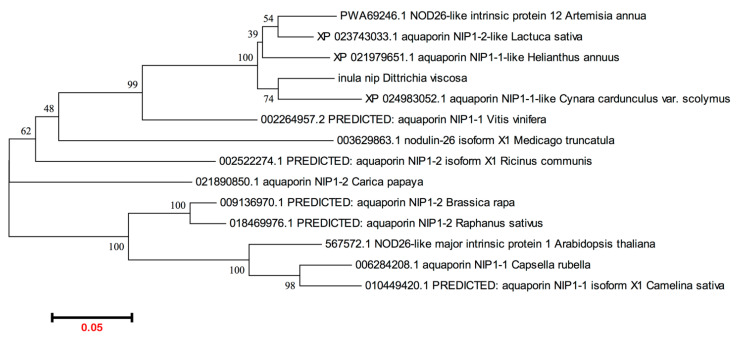
Phylogenetic analysis of plant NIPs1. The sequences were retrieved from NCBI genbank: >inula nip (*Dittrichia viscosa*); >PWA69246.1 NOD26-like intrinsic protein 1,2 (*Artemisia annua*); >XP_023743033.1 aquaporin NIP1-2-like (*Lactuca sativa*); >XP_024983052.1 aquaporin NIP1-1-like (*Cynara cardunculus var. scolymus*); >XP_021979651.1 aquaporin NIP1-1-like (*Helianthus annuus*); >567572.1 NOD26-like major intrinsic protein 1 (*Arabidopsis thaliana*); >006284208.1 aquaporin NIP1-1 (*Capsella rubella*); >010449420.1 PREDICTED: aquaporin NIP1-1 isoform X1 (*Camelina sativa*); >009136970.1 PREDICTED: aquaporin NIP1-2 (*Brassica rapa*); >018469976.1 PREDICTED: aquaporin NIP1-2 (*Raphanus sativus*); >021890850.1 aquaporin NIP1-2 (*Carica papaya*); >002522274.1 PREDICTED: aquaporin NIP1-2 isoform X1 (*Ricinus communis*); >002264957.2 PREDICTED: aquaporin NIP1-1 (*Vitis vinifera*); >003629863.1 nodulin-26 isoform X1 (*Medicago truncatula*). The program MEGA6.0 was used.

**Figure 3 plants-11-01968-f003:**
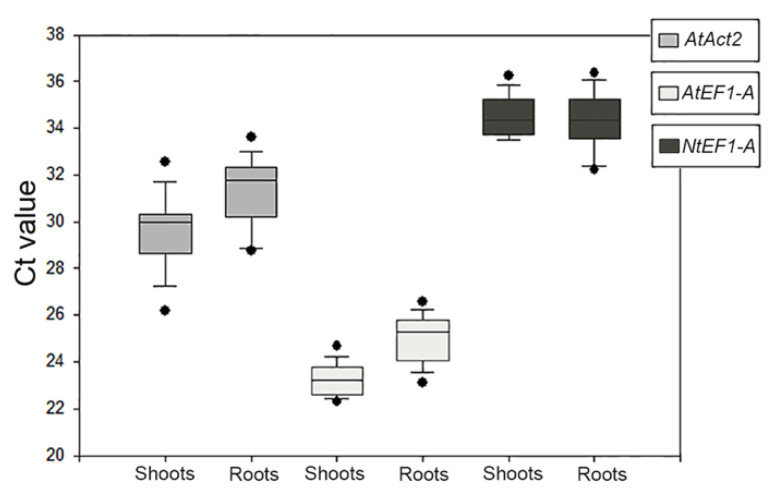
Quantitative real-time polymerase chain reaction of raw Ct values of the three amplification products given by the primer pairs used to identify the reference genes for *Dittrichia* in all experimental conditions in each sample (shoots and roots). The results are presented with box plots (middle bar, median; box limits, upper and lower quartiles; whiskers, min. and max. values).

**Figure 4 plants-11-01968-f004:**
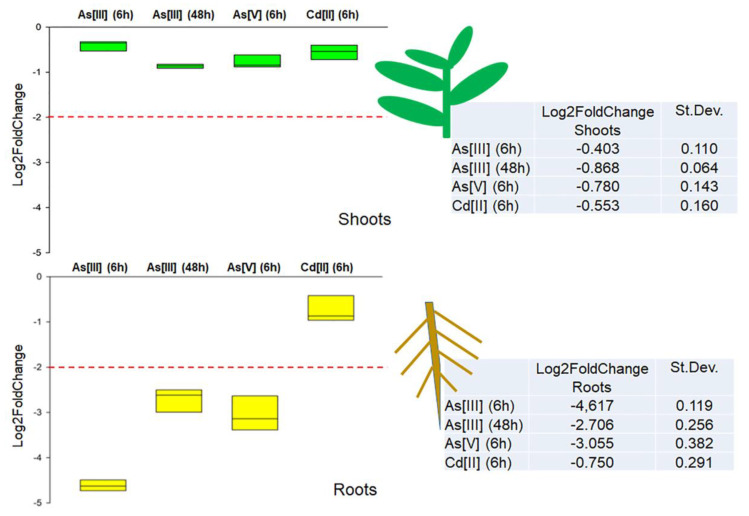
Quantitative real-time polymerase chain reaction of *DvNip1.1* in shoots and roots of DI3-F plants subjected to As[III] (6 h), As[III] (48 h), As[V] (6 h), and Cd[II] (6 h). The expression of *DvNip1.1* gene is reported as transcript inhibition level (log2 of fold change) with respect to control (without HM). The results of three independent biological and three technical replicates are presented with box plots (middle bar, median; box limits, upper and lower quartiles; whiskers, min. and max. values).

**Figure 5 plants-11-01968-f005:**
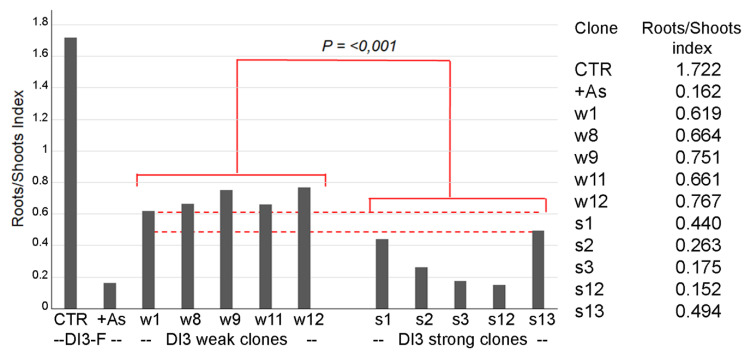
Quantitative real-time polymerase chain reaction of *DvNip1.1* in two groups of four plants showing the weakest (weak) or the strongest phenotype (strong). Amplification output values are expressed as 2−∆Cq ± SD for *DvNip1.1* mRNAs (in control condition and after HM stress) and are considered as proportional to the amount of mRNA target according to [27]. The results were showed as roots/shoots index, given by the ratio of the value of *DvNip1.1* root 2−∆Cq on the value of *DvNip1.1* shoot 2−∆Cq. Data were submitted to one-way analysis of variance (ANOVA), and differences among two group were detected using comparison procedures within each group (Holm–Sidak test, *p* < 0.001).

**Table 1 plants-11-01968-t001:** Primer nucleotide sequences.

Primer	Sequence	Application
DvNipDegR	CCYAARCTYCTTSCYGGRTTCAT	gPCR
DvNipDegF	GGYGCHCATTTYAAYCCDGC	gPCR
DvNip3′GSP1-F	AACCCTGCCGTCACCATTG	3′RACE
RACE (AUAP)	GGCCACGCGTCGACTAGTAC	3′-5′RACE
DvNip5′GSP2R(nested)	CAATGGTGACGGCAGGGTTG	5′RACE
DvNip5′GSP1-R	GAAACCTACTGCAGGATGC	5′RACE
DvNip-Full-F	ATGGCGGAGATAGAGGGAG	RT-PCR
DvNip-Full-R	TCATCGTTTAGAACTATTACG	RT-PCR
AtEF1Afor	AGCCCAAGAGGCCATCAGA	RT-qPCR
AtEF1Arev	CCACTGGCACCGTTCCA	RT-qPCR
AtAct2for	TACAGTGTCTGGATCGGTGGTT	RT-qPCR
AtAct2rev	CGGCCTTGGAGATCCACAT	RT-qPCR
AtAct8for	GCTGGATTCGCTGGAGATGA	RT-qPCR
AtAct8rev	CATGATGTCTAGGTCGACCAACA	RT-qPCR
NtEF1Afor	TGAGATGCACCACGAAGCTC	RT-qPCR
NtEf1Arev	CCAACATTGTCACCAGGAAGTG	RT-qPCR
DvNip1.1for	TTTTGGTGCGTGGGTCTACA	RT-qPCR
DvNip1.1rev	ACGACTGCTAGCATACCGAA	RT-qPCR
HaEF1Afor	TCAACCAACCTCGACTGGTA	PCR
HaEF1Arev	TCAACGCTCTTGATGACACC	PCR

## Data Availability

All data available in manuscript.

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
