# Peer review of "Evaluation of Dittrichia viscosa Aquaporin Nip1.1 Gene as Marker for Arsenic-Tolerant Plant Selection"

_plants, 2022, doi:10.3390/plants11151968_

Round 1

Reviewer 1 Report

The authors elucidated the role of NIP1.1 in relation to arsenic tolerance in Dittrichia viscosa. However, as mentioned in the Introduction, there is not enough explanation as to why NIP1.1 was chosen among the many AQPs. Since there are many papers on AQPs related to arsenic tolerance in Arabidopsis, it would be nice to add a discussion on the possible role of other AQPs related to arsenic tolerance in D. viscosa. I added PDF file for need minor change. 

Author Response

We thank the reviewer for the improvement of our manuscript and for the valuable comment about the focusing of NIP1.1 selection to justify this study.

To better introduce the selection of NIP1.1 for our study, we added at Pag.2 (of 15)  the following text including a new reference:

"As a phosphate analog As(V) enters plant cells through phosphate transporters (PHTs) while As(III) and methylated As species enter the cells and get translocated or detoxicated by aquaporins and several additional transporters (Mondal et al. 2022). Despite this complex scenario, it was shown that tolerance to As is not related to a reduced uptake but it is increased by the mutation of the aquaporin AtNip1.1 (Kamiya et al., 2009; Barozzi et al., 2019). …"

In addition we inserted a conclusive sentence at page 9 (of 15), in the discussion section:

"This observation is important to identify a marker for As tolerance considering that other channels and transporters are related to As and its uptake. Even among AQPs AtNIP5.1, OsPIP2;4, OsPIP2;6 and OsPIP2;7 have all shown to influence As[III] uptake and tolerance (Mondal et al., 2022)."

We did not add further specific references on aquaporins related to Arsenic because our focus on NIP1.1 depends, as stated, from the observation of the acquired tolerance after NIP1.1 mutation in Arabidospis.

If we misunderstood the reviewer concern, we will be happy to expand literature with relevant publications.

Reviewer 2 Report

Heavy metal pollution control and plant tolerance is a very meaningful topic. This study closely follows the current research hotspot and has important practical research value.  This article can be published after modification.

1, The literature on the sources of arsenic pollution is too old (1973 and 2001), and now the sources of arsenic pollution have changed greatly.

2, Writing needs to be careful. For example, the title number: the Result is 3, and the Discussion is 3; there is also the problem of superscripts and subscripts in the text.

Author Response

We thank the reviewer for the appreciation and the valuable comments.

We updated the background literature about As pollution with two 2021 references that confirm the concept we wanted to support.

We also performed a general editing revision, for example in chemical formulas, and we hope to have found all the typos in the text mentioned by reviewers.